# The Entity of Cerebellar Mutism Syndrome: A Narrative Review Centered on the Etiology, Diagnostics, Prevention, and Therapeutic Options

**DOI:** 10.3390/children10010083

**Published:** 2022-12-30

**Authors:** Dimitrios Panagopoulos, Georgios Stranjalis, Maria Gavra, Efstathios Boviatsis, Stefanos Korfias, Ploutarchos Karydakis, Marios Themistocleous

**Affiliations:** 1Neurosurgical Department, Pediatric Hospital of Athens, ‘Agia Sophia’, 45701 Athens, Greece; 21st University Neurosurgical Department, ‘Evangelismos’ Hospital, University of Athens, Neurosurgery, Medical School, 10676 Athens, Greece; 3Radiology Department, Pediatric Hospital of Athens, ‘Agia Sophia’, 45701 Athens, Greece; 42nd University Neurosurgical Department, ‘Attikon’ Hospital, University of Athens, Neurosurgery, Medical School, 12462 Athens, Greece; 5Neurosurgical Department, General Hospital of Athens ‘Gennimatas’, 11527 Athens, Greece

**Keywords:** cerebellar mutism, vermis, medulloblastoma, dentate nucleus, occupational/speech therapy

## Abstract

Cerebellar mutism syndrome (CMS), also known as posterior fossa syndrome, is an entity that entails a constellation of signs and symptoms which are recorded in a limited number of pediatric patients who have been operated on mainly for tumors involving the posterior cranial fossa, and more precisely, the region of the vermis. Medulloblastoma seems to constitute the most commonly recognized pathological substrate, associated with this entity. The most prevalent constituents of this syndrome are noted to be a, often transient, although protracted, language impairment, emotional lability, along with cerebellar and brainstem dysfunction. Apart from that, a definite proportion of involved individuals are affected by irreversible neurological defects and long-lasting neurocognitive impairment. A bulk of literature and evidence based on clinical trials exist, which reflect the continuous effort of the scientific community to highlight all perspectives of this complex phenomenon. There are several circumstances that intervene in our effort to delineate the divergent parameters that constitute the spectrum of this syndrome. In summary, this is implicated by the fact that inconsistent nomenclature, poorly defined diagnostic criteria, and uncertainty regarding risk factors and etiology are all constituents of a non-well-investigated syndrome. Currently, a preliminary consensus exists about the identification of a group of diagnostic prerequisites that are managed as sine qua non, in our aim to document the diagnosis of CMS. These include language impairment and emotional lability, as proposed by the international Board of the Posterior Fossa Society in their consensus statement. It is common concept that midline tumor location, diagnosis of medulloblastoma, younger age at diagnosis, and preoperatively established language impairment should be accepted as the most determinant predisposing conditions for the establishment of this syndrome. A well-recognized pathophysiological explanation of CMS includes disruption of the cerebellar outflow tracts, the cerebellar nuclei, and their efferent projections through the superior cerebellar peduncle. Despite the relative advancement that is recorded regarding the diagnostic section of this disease, no corresponding encouraging results are reported, regarding the available treatment options. On the contrary, it is mainly targeted toward the symptomatic relief of the affected individuals. The basic tenet of our review is centered on the presentation of a report that is dedicated to the definition of CMS etiology, diagnosis, risk factors, clinical presentation, and clinical management. Apart from that, an effort is made that attempts to elucidate the paramount priorities of the scientific forum, which are directed toward the expansion our knowledge in the era of diagnostics, prevention, and therapeutic options for patients suffering from CM, or who are at risk for development of this syndrome.

## 1. Introduction

The term cerebellar mutism syndrome is universally utilized in order to express the loss, transient or permanent, of speech that is intimately related to any kind of cerebellar insult. The historically first report of this term (CMS) is encountered in 1958, referring to a pediatric patient who was unable to speak after resection of a space-occupying lesion that was involving the posterior fossa. This entity was termed akinetic mutism [1,2]. The first relevant cases of postoperative mutism affecting the pediatric population, which were an operation involving the structures of the posterior fossa, were recorded in the 1970s [3,4]. Additionally, in 1984, Wisoff and Epstein shared their clinical knowledge, centered on pediatric patients who developed delayed onset cranial nerve palsies, emotional lability, and speech impairment. They collectively named this group of deficits as pseudobulbar palsy [5]. After that, in 1985 Rekate et al. presented the first small case series that was attempting to describe this entity. A manuscript was published that focused on six cases of cerebellar mutism that occurred after an operation involving the posterior cranial fossa [6]. Kirk et al. utilized the term posterior fossa syndrome in order to describe the same combination of clinical and neurological findings that was attributed to the term cerebellar mutism syndrome [7]. After this initial demarcation of that pathological entity, over 400 cases have been recorded in the literature [8]. In the majority of cases, cerebellar mutism is not observed as an isolated neurological finding, but, instead, it is a component of a more complex constellation of neurological deficits. More precisely, it accompanies a widespread cluster of neurological, emotional, and behavioral disturbances. Based on the complexity of this entity, as well as on the fact that it incorporates multiple independent signs and symptoms, the term cerebellar mutism is frequently encountered in the existent literature as cerebellar mutism syndrome or posterior fossa syndrome. According to a recently published paper [1], this syndrome is determined by the simultaneous occurrence of symptoms, which include mutism/reduced speech, emotional lability, cerebellar syndrome, brainstem dysfunction, hypotonia, and oropharyngeal dysfunction/dysphagia. There is a wide consensus regarding this definition of the syndrome [9,10,11,12,13].

Regarding the most common conditions that accompany this syndrome, it is widely accepted that the most common scenario refers to children that have undergone an operation for a pathologic condition that affects the posterior cranial fossa. Nevertheless, it is not restricted to this patient population, and it may accompany other pathologic conditions that are affecting the cerebellum, either in adults or in children [14,15]. In childhood, the greatest proportion of solid space-occupying lesions involves the central nervous system and, more specifically, the cerebellum (approximately 80% of them) [16]. Mutism is considered a severe and devastating adverse neurological sequelae of neurosurgical approaches centered on the resection of tumors of the posterior fossa, and its relative prevalence could not be underestimated. Its clinical course is largely unpredictable, with the spontaneous resolution being the most frequent outcome. Nevertheless, this is not true for the other deficits that constitute this entity. Namely, dysarthria, cognitive, and behavioral disturbances as well as language disorders that are evident during the mute phase are not always adequately resolved, and thus constitute a disabling condition for the patient.

The exact prevalence of cerebellar mutism syndrome after operations that involve the region of the posterior fossa in the pediatric population remains to be specified, probably because the recognition of relevant cases underestimates their true prevalence as the diagnostic criteria are not strict and universally adopted. Nevertheless, it ranges between 11% and 29% [8], even though more recent prospective studies that incorporated larger groups of pediatric patients reported an incidence of 27.7% [17] and 24% [11] respectively. The patients’ mean age at presentation was 6–7 years, based on those reports. Nevertheless, whenever sex and patient age were investigated as possible independent predisposing conditions for the establishment of the syndrome, no statistically significant correlation or differentiation could be established. On the contrary, the histopathology of the offending lesion was recognized as a potential determining factor of the risk for the establishment of the syndrome. Namely, medulloblastoma resection indicated a higher correlation rate (40%), in comparison with pilocytic astrocytoma (16%) or ependymoma (4%) [12,17,18,19,20,21]. There is significant variability regarding the estimated rate of occurrence of this syndrome after an operation on the posterior cranial fossa. This could be attributed, at least in part, to the absence of strict definition criteria for the syndrome [9,22,23]. Another important parameter of this syndrome is related to the determination of the most important risk factors that have an impact on the possibility of the establishment of relevant neurological deficits. Namely, midline tumor location and medulloblastoma histologic variant [24,25,26,27,28], younger patient age [9], left-handedness, aggressive resection, infiltration of the brainstem, and tumor diameter > 5 cm have been implicated accordingly.

Another implicating factor was considered to be the existence of pre-surgical language deficiencies, although this association was based on a smaller patient population [29]. Although the entity of cerebellar mutism syndrome is inherently related to tumor lesions of the cerebellum, sporadically cases have been reported that implicated other pathologic conditions as precipitating factors. These included trauma [30], stroke [31], and inflammation [32].

## 2. Methods

### 2.1. Search Strategy

We executed a specific term-related search via the aid of the Thomson Reuters Web of Science database, in order to recognize the most highly cited articles related to brain metastasis, until December 2022. Our query term was: “cerebellar mutism syndrome” OR “post-operative cerebellar mutism syndrome” OR “posterior fossa syndrome”. Moreover, no refinement of the results by using restriction criteria, such as publication dates was performed.

### 2.2. Clinical Presentation and Time Course

The cerebellar mutism syndrome is chronically evolving into three consequent, succesive phases. More precisely, cerebellar mutism is not evident after the initial recovery of the patient from the operation. Instead of that, there is a time interval between the operation and the establishment of the syndrome, which varies from a few hours up to several days following the operation [11,33]. Moreover, mutism is always a transient disability with unpredictable duration, with reports depicting a time range between a few days to several months [14]. The resolution of mutism is not spontaneous and follows a gradual process. As is already mentioned, after the mute phase, several parameters of the syndrome continue to be evident, to various extent and in different combinations (motor speech and language deficits, cognitive, emotional and behavioral disorders).

During the mutistic phase, high-pitched crying is the only clinical sign that is relevant with vocalization [11]. During this phase, a constellation of neurological signs may be evident, indicating the existence of cerebellar and/or brainstem injury. These may include ataxia, involuntary eyelid closure, pyramidal tract signs, horizontal gaze paralysis, cranial nerve palsies, and oropharyngeal dyspraxia [7]. The spectrum of this syndrome incorporates behavioral and emotional disturbances, which may manifest with emotional lability, apathy, and autistic-like behavior [34,35]. Bizarre personality changes may be evident as forced laughing or crying.

The dissolution of the components of mutism does not follow a universal pattern but, instead, can follow one of several different alternative models [36,37]. The most commonly encountered patterns include dysarthria, which is not accompanied by any signs of higher language dysfunction [38]. Alternatively, a language disorder without dysarthria may be recorded. Another clinical scenario is that behavioral disturbances constitute the next manifestation of this syndrome. However, this concept is not universally accepted, as some researchers consider that dysarthria is a common characteristic that is evident in virtually every patient who is recovering from the acute phase of the syndrome [39]. The existence of long-term neurological deficits, as well as persistent neurocognitive impairment, has been widely accepted as constituents of that entity [18,40,41]. Differences in evaluation criteria and subgroups of selected patients could be the incriminating factor, capable to interpret the divergent non-consistent findings of reported trials. It would be beneficial to adopt universally accepted methods, centered on the evaluation of speech and language outcomes for upcoming studies. Because of that, the Posterior Fossa Society made an attempt to define this syndrome as a ‘postoperative pediatric cerebellar mutism syndrome, characterized by delayed onset mutism/reduced speech and emotional lability, commonly accompanied by hypotonia, oropharyngeal dysfunction, cerebellar motor deficits, cerebellar cognitive affective syndrome, and brain stem dysfunction [12].

Attempting to define the specific components of dysarthria attributed to cerebellar mutism syndrome, the most commonly reported features include the existence of reduced speech rate and the utilization of short but grammatically correct speech. In addition, there is a report [37] that was centered on two children that had a telegraphic language in the post-mutistic phase.

Regarding the behavioral changes that characterize the post-mutistic phase, these share a lot in common with autism. It is a common concept that the affected children have a decreased ability to be part of a team with their colleagues and avoids physical and eye contact. Another important remark is that speech is devoid of emotional inflections and is seldom utilized as a means of communication [37]. Emotional disturbances are frequently present and consist of lability and irritability [36,37]. Regarding the long-term constituents of cerebellar mutism syndrome, they consist of ataxia, speech or language dysfunction, including dysarthria and dysfluent/slower speech, and intellectual impairment [11].

As already mentioned, the Posterior Fossa Society held an international consensus meeting and defined the discrete aspects of this post-operative syndrome. The condition is typically viewed as a pediatric syndrome, the core features of which are: (1) mutism occurs after resection of a cerebellar mass lesion; (2) there is generally a delayed onset of speech loss after a brief interval of 1–2 days of normal speech post-surgery; (3) mutism is transient and generally lasts from 1 day to 6 months; (4) mutism is followed by severe dysarthria which usually recovers favorably in 1–6 months but may persist in some cases; (5) there are frequent associations with other neurological disturbances, such as long tract signs and neurobehavioral abnormalities.

On the contrary, several exclusion criteria have been adopted. More precisely, children who never had been mute after surgery to the posterior cranial fossa were not included in the analysis. In order to ensure group homogeneity, mute children who were not were also not considered as able to be incorporated in that group. Moreover, children with mutism that was related with atraumatic or infectious origin, along with cases of mutism that are established after brainstem surgery are excluded. The main reason for that is related to the fact that there exists a significant possibility that diffuse injury, that is cable of extending beyond the boundaries of cerebellum, to accompany these cases.

## 3. Discussion

### 3.1. Possible Causes of Cerebellar Mutism

The fact that this syndrome is accompanied by a wide, and divergent, spectrum of clinical manifestations, which come to clinical attention at different time periods, is an obstacle to the establishment of a universal anatomic-pathophysiological circuit, capable to interpret this entity. Several proposals have been adopted, hypothesizing that local tissue damage of the cerebellum and brainstem should be implicated in the development of the syndrome. Apart from that, dysfunction of regions of the cerebral cortex, because of damage to cerebello-cortical pathways, should be taken into consideration. Another important factor that may be of relevant importance is the discrimination of the offending pathological substrates under the terms permanent and transient (e.g., edema). The proposed offending mechanisms should not be considered contradictory. On the contrary, they may share different and distinct roles in the pathophysiology of cerebellar mutism syndrome, acting at individual time points.

The current concept regarding the pathophysiologic explanation of the clinical features of the syndrome that occur during its distinct phases could be summarized under the following statements:

Mutism itself is inherently related to supratentorial dysfunction, mediated by crossed cerebello-cerebral diaschisis. This refers to a condition that is characterized by an asymmetry of blood flow or metabolism in supratentorial structures contralateral to a remote cerebellar lesion [42,43,44,45,46,47,48,49]. Injury of the dentato-thalamocortical pathway is suggested to be the offending pathology, intimately related with the crossed cerebello-cerebral diaschisis [49,50]. There are reports that support the existence of an association between the damage to the frontal cortex and mutism [45,46,47,48]. Another fact that supports that concept is the existence of reports which correlate behavioral disturbances in the pediatric population with cerebellar mutism with frontal cortex dysfunction [34,49,50,51]. Based on that evidence, we could state that the initial phase of symptoms relevant to cerebellar mutism is primarily attributed to cerebral cortical dysfunction, caused by crossed cerebello-cerebral diaschisis.

Regarding dysarthria, its anatomic substrate has been proposed to be damage of the dentate and interposed nuclei, as well as in the cerebellar cortex, with lesions located on the paravermal lobule VI [52,53,54]. This is supported by the intimate anatomical relationship of cerebellar nuclei to the cerebellar midline, rendering them vulnerable to (permanent or transient) lesioning during tumor resection.

A distinct role of the cerebellum in language function, which is not limited to motor speech articulation, has been extensively investigated [55]. Agrammatic speech was the most frequently encountered component of aphasia of cerebellar origin· the posterolateral hemispheric region, along with the adjacent compartments of the dentate nuclei are considered to contribute to the linguistic process.

Finally, intra-operative tissue damage to the region of the vermis is reported to share an intimate relationship with persisting affective disturbances as part of the ‘cerebellar cognitive affective syndrome’ [56]. Apart from that, a lot of researchers have developed imaging-based predictive models in order to elucidate the underlying pathophysiology of CMS [26,57].

### 3.2. Neuroimaging and CMS

The DTC pathway represents an important outflow tract from the cerebellar nuclei towards the cerebral cortex, as it connects the dentate nucleus, via the contralateral red nucleus and thalamus, to the contralateral cerebral cortex There is consensus that interruption of this pathway constitutes the main pathological substrate for CMS [8,58]. Abnormal signal intensities in regions that involve the proximal efferent cerebellar pathway, the middle cerebellar peduncle, and the vermis have repeatedly been reported [35,38,42,59,60,61]. A recent survey identified that lesions that are located along with the cerebellar outflow could be considered predictors for the development of CMS [62].

The existence of postoperative vermian lesions should not be considered as equivalent to the development of CMS. Several experts state that the recognition of diffusion abnormality in the region of the vermis is not assumed as mandatory for the appearance of CMS [42,63]. Another useful sequence that is capable of assessing the integrity of white matter tracts is diffusion tensor imaging, and this has been utilized in order to assess the integrity of the DTC pathway in cases of established CMS.

Intraoperative MRI could be considered a sufficient adjunct in order to verify the existence of MRI abnormalities that take place during or immediately after the operation. More precisely, diffusion-weighted imaging [64] can verify the existence of vasogenic or cytotoxic edema, intimately related to the surgical approach.

### 3.3. Pathophysiology and Anatomy

Albeit numerous hypotheses have been proposed, centered on the pathogenesis of POPCMS (post-operative cerebellar mutism syndrome), a detailed delineation of the underlying pathologic substrate needs to be performed. The development of this syndrome is intimately related to several anatomical structures that are located in the anatomical territory of infratentorial and supratentorial compartments. This fact has led to the suggestion that a variety of circuits are implicated, as well as domes that are located both in the vicinity of the surgical field and at a distance from that, are responsible for its appearance [65,66].

An intraoperative surgical lesion located to the relevant anatomical substrates, more precisely the pECP (proximal efferent cerebellar pathway), has been widely accepted as a precipitating factor for the establishment of POPCMS. A common immediate post-operative imaging finding is the establishment of cerebral edema in the vicinity of the resection cavity, which could be vasogenic or cytotoxic in origin. Post-operative edema reaches its maximum density at about 24 h after the operation, remains relatively consistent for the next 3 days, and is gradually disappeared at post-operative day 7 [67]. The importance of that remark is enhanced by several radiological studies, which have established the existence of a definitive correlation between edema in the pECP domes and the establishment of POPCMS [35,38,68]. The availability of intra-operative MRI has led to the admission that the edema should be intimately correlated with surgical interventions and not with other pathophysiological interactions [69]. All these references suggest the implication of direct surgical maneuvers in the region of the pECP as an offending mechanism for POPCMS, along with the direct axonal injury that is inevitable sequelae of the surgical approach [70,71].

Another implicated mechanism includes the tissue damage that is provoked by the thermal injury that is related to the operative procedure [58]. Researchers mention that the increased heat which is associated with aspiration of tumor tissue via the use of CUSA could potentially be the offending mechanism of the tissue damage to the brain parenchyma which surrounds the lesion, more precisely the pECP structures [72,73,74,75]. The anatomical distribution of this insult shares a lot in common with the distribution of the abnormal signal patterns that are delineated on the DWI sequence in individuals suffering from POPCMS. In accordance with that hypothesis, it has been mentioned that the restricted use of CUSA, in combination with avoidance of excessive retraction, as well as the judicious utilization of electrophysiological monitoring, is an effective means, aiming toward the reduction of the incidence of POPCMS [68].

### 3.4. Anatomical Substrate of CMS

It is almost universally accepted that damage that involves the DTC pathway constitutes the main anatomical substrate of CMS [76,77]. Several different pathophysiologic mechanisms have been implicated in the context of DTC disruption, namely cerebral cerebellar diaschisis, edema, perfusion deficit, and cerebellar vermis injury. A brief description of the aforementioned mechanisms follows.

### 3.5. Cerebral Cerebellar Diaschisis

This phenomenon is described as the functional deficit that refers to a definitive region of the brain and is causally related to damage that has occurred to another, remote, brain region. The net effect of such an insult is the generation of an excitatory input to the inhibited area [42]. The disruption of these pathways leads to the loss of excitatory input from the cerebellum to the relevant recipient cerebral cortical areas. These include the motor, premotor, and prefrontal regions, which are known to interfere with the functions that are affected by cerebellar mutism and result in their loss [8,42,78]. Although diaschisis was initially encountered as a transient phenomenon, recent evidence has shown that it could be related to long-term damage, which was affecting the associated remote parenchymal brain areas [79]. This remark offers the substrate in order to justify the language and cognitive defects that were registered in a great percentage of patients suffering from CMS, even 1 year after the establishment of its diagnosis [11,38]. There seems to be a time delay between the establishment of remote hypoperfusion from the onset of symptoms, which displays a significant variation among different studies [80,81]. Nevertheless, this observation may serve as an interpretation for the delayed onset of CMS symptoms.

### 3.6. Edema

The development of postoperative edema is an additional suggested mechanism, mainly due to the fact that its appearance and evolution follow a parallel time course with the relatively delayed onset of CMS [60]. Supportive evidence on that is derived from diffusion tensor imaging studies, which have definitively stated the correlation between edema and the development of post-operative mutism, located in the superior cerebellar peduncles, the pons, and mesencephalon [35,60]. Nevertheless, a drawback of this proposed mechanism is related to its inability to provide an explanation about the fact that mutism does not subside after the resolution of the postoperative edema, whereas a group of symptoms may persist lifelong [35].

### 3.7. Perfusion Deficit

This theory provides an answer to our question regarding the delayed onset of CMS. According to that, it may be causally related to the development of perfusion deficit. The establishment of postoperative vasospasm could provide a meaningful explanation for the delay in the onset of CMS. Apart from that, the transient ischemia that eventually follows vasospasm is in accordance with the clinical establishment and potentially delayed remission of CMS that occurs with the re-establishment of blood flow [71,82].

### 3.8. Cerebellar Vermis Injury

Destruction of tissue structures that refers to the cerebellar vermis and is related to the surgical approach to the offending lesion has been recognized to be of critical importance in the development of CMS [77]. We have identified connections of the vermis with the cerebellar nuclei that are believed to constitute a major contributor to the establishment of fluent speech [77,83]. The trans-vermian approach, which involves splitting of the vermis in order to surgically approach tumors in the midline of the posterior fossa, has been proposed as a potential contributing factor, albeit its contribution is not completely elucidated [77,84,85].

### 3.9. Prevention and Treatment

Although initially, most experts considered that dysarthria that was attributed to post-operative CMS could recover in a short-term fashion without any residual deficits [38,86], this concept was not confirmed by subsequent studies. More precisely, it was realized that a wide spectrum of permanent defects was observed, the most evident of which was persistent dysarthria, language impairment, and dysphagia [11,87,88]. De Smet et al. [89] and Huber et al. [90] conducted studies centered on the long-term course of dysarthria encountered under the term CMS. More precisely, Smet stated that all pediatric patients suffering from pCMS manifested dysarthria in the early time-course of CMS, whereas in 91.7% of them, persistent motor speech deficits were recorded up to 12 years after surgery. Their recorded data were in accordance with other findings, which reported persistent motor speech defects in the long-term, and approved the standpoint that CMS should be considered as a prognostic factor for long-term dysarthria in children operated for cerebellar tumors [29,87].

Another important aspect of this issue is the identification of the most important risk factors that are crucial for the establishment of CMS. More precisely, according to Di Rocco et al. [29], the existence of pre-operative language impairment was considered an important risk factor for the development of CMS. They demonstrated that, when they examined patients with posterior fossa tumors, even preoperatively, impairment of selective speech and language functions may be evident. More precisely, there are only limited data published in the literature focused on this issue [91,92,93]. Apart from the results published by Di Rocco et al., several other articles simply mentioned the existence of preoperative language impairment as a potential risk factor for the establishment of CMS postoperatively [10,11,15,37,48,65,76,78]. On the contrary, only Beckwitt–Turkel evaluated extensively the significance of preoperative language impairment, and conducted similar conclusions [94]. According to a recent report [42], 28.5% of children were suffering from preoperative language impairment and developed cerebellar mutism postoperatively. Moreover, when these patients were followed up for a protracted time period, it appeared that the complete resolution of CMS occurred after a protracted time period.

Another issue that is of great clinical significance relates to the rehabilitation of speech and language deficits that were established during and after the mute phase. Currently, no treatment modality centered on the speech and language deficit of CMS is available. Nevertheless, several efforts have been performed in order to ameliorate the neurological deficits that are associated with mutism in its acute stage, based on pharmaceutical therapeutic regimens. Namely, corticosteroids, fluoxetine, thyrotropin-releasing hormone, bromocriptine, midazolam, and zolpidem have been utilized, albeit their efficacy is not verified at all [95].

Despite an effective and widely accepted therapeutic protocol for these children lacking, timely intervention for children harboring tumors of the posterior fossa has been considered an effective means to minimize speech and language deficits [96]. Based on current data, it seems that rehabilitation of speech and language in children suffering from CMS necessitates a multi-phase implementation of evidence-based guidelines and recommendations that specify and underline the most significant risk factors, the registered specific deficits, and clinical data that are centered on the evolution of the syndrome over time [97,98,99]. Another issue that has extensively been studied in the literature is centered on the long-term neurocognitive outcomes of children suffering from CMS. According to most studies, it seems that these children demonstrate a wide spectrum of neurocognitive and neuroemotional deficits during their follow-up after their operative treatment [11,68].

### 3.10. Minimizing the Risk of CMS

A systematic effort directed toward the elimination of the possibility of the development of CMS should be oriented on the comprehensive determination of the implicated predisposing factors, either preoperative or intraoperative [84]. Attempting to develop a preoperative evaluation scale to stratify the potential risk, Walker et al. [26] introduced a model which included six relevant factors. These included primary tumor location, as it was specified by MRI, bilateral middle cerebellar peduncle involvement (invasion and/or compression), dentate nucleus invasion, and age at surgery > 12.4 years. The ability to accurately predict that risk based on data that could be collected preoperatively has a great impact on the determination of our surgical plan, namely the selection of the safest approach and the extent of anticipated resection.

Another important issue that still constitutes a matter of considerable debate relates to the selection of the safest surgical approach, which is a comparison between a telovelar versus a trans-vermian surgical corridor. A lot of studies exist that have proposed that a split-vermis approach may be related to a substantial risk of CMS [100,101,102,103]. On the contrary, other authors have noted that the avoidance of splitting of the vermis did not have any significant impact on the development of CMS [104]. Although the telovelar approach has been considered an alternative surgical option that is beneficial in terms of avoiding CMS, its real advantage remains questionable. According to a recent review, the combined selection of a telovelar approach, the restricted use of CUSA, as well as the avoidance of intraoperative retraction constitutes the most effective method for the prevention of the development of CMS. Our tenet should be to refine our surgical strategy and investigate the majority of available measures in order to implement a therapeutic protocol capable of essentially minimizing the overall prevalence of CMS.

### 3.11. Treatment and Rehabilitation of Behavioral and Cognitive Problems in the Context of CMS

Children who fulfill the criteria of CMS are at increased risk for significant long-term cognitive and psychosocial morbidity. This fact makes it necessary to form a group of patients for whom interventions and rehabilitation should constitute the main priority, except for interventions that focus exclusively on motor and speech functions. The primary areas of rehabilitation therapy that these children receive target the motoric and speech/language deficits that are established in the acute period after the initiation of pCMS and this is achieved via physical, occupational, and speech/language therapies. A lot of patients are not implemented in the acute phases of pCMS with cognitive interventions. This is mainly related to the fact that cognitive issues are recognized later on, during a later stage of CMS.

### 3.12. Cognitive Rehabilitation

Research on cognitive remediation programs is mainly focused on two distinct categories: (1) Face-to-face therapeutic sessions targeting specific functions such as executive function, attention, memory, and academic achievement, and (2) computer-based intervention programs that are implemented within the home setting. The net result of these interventions is an improvement in visual working memory and parent-reported learning difficulties. Notably, the most noticeable improvement was recorded in children with higher intellectual functions, recorded before the application of any treatment [86,105]. A table with major findings of previous case series/studies on CMS follows (Table 1).

## 4. Conclusions

Cerebellar mutism constitutes a considerable and possibly underestimated complication in a relatively large number of children that underwent a posterior fossa surgery for tumor resection, especially when it is located in the midline. It self-subsides on its own but is frequently associated with long-term speech deficits and other neurocognitive deficits. Preoperative tumor infiltration into the brainstem, as well as evidence of post-operative insult of the bilateral dentato-thalamocortical tract, are currently considered the major determinant factors for the establishment of this syndrome. When the implicated pathophysiological substrate for this syndrome is considered, dysfunction of the frontal cortex, mediated by crossed cerebello-cerebral diaschisis, is the presumed primary factor. There is a lack of definitive evidence centered on the treatment of this syndrome, and this, at least, may be attributed to the fact that there is a lack of any controlled studies directed toward the treatment or prevention of cerebellar mutism.

Another important issue of this entity is that it is often accompanied by long-term neurological symptoms, as well as neurocognitive deficits that persist throughout life. These defects adversely affect the overall quality of life and pose significant restrictions to the patients’ ability to carry out activities of daily living, imposing great obstacles to patients and their families. It is of utmost importance to obtain a deeper knowledge of all aspects of this syndrome and the most effective way to achieve this goal is through a concentrated effort to formalize the criteria needed for the designation of the syndrome, specify the most relevant clinical predictors and outcomes, and recognize the pathophysiologic substrate and an effective management protocol for CMS.

## Figures and Tables

**Table 1 children-10-00083-t001:** Pivot table, incorporating all relevant major findings of previous case series/studies, relevant with CMS.

Major Findings	Relevant Case Series/Studies
Interruption of the dentato-thalamo-cortical (DTC) pathway is the main cause for CMS.	Avula S. Radiology of post-operative pediatric cerebellar mutism syndrome [58] https://doi.org/10.1007/s00381-019-04224-x
Direct cerebellar injury is the likely reason for persisting deficits after the mute period.	Küper et. al. Cerebellar mutism [15] https://doi.org/10.1016/j.bandl.2013.01.001
Restoration of motor functions and communication relies heavily upon physiotherapy and occupational, speech, and language therapy	Paquier et. al. Post-operative cerebellar mutism syndrome: rehabilitation issues [86] https://doi.org/10.1007/s00381-019-04229-6
The appearance of complex dysarthria in the postoperative period is a negative prognostic factor for the long-term persistence of speech disturbances	Bianchi et al. Cerebellar mutism: the predictive role of preoperative language evaluation [92] https://doi.org/10.1007/s00381-019-04252-7
Use of a telovelar over a transvermian approach, avoidance of the CUSA, and minimization of heavy retraction during surgery reduce the incidence and severity of CMS	Cobourn et al. Cerebellar mutism syndrome: current approaches to minimize risk for CMS [84] https://doi.org/10.1007/s00381-019-04240-x

## Data Availability

Data is contained within the article.

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
