# Peer review of "The Entity of Cerebellar Mutism Syndrome: A Narrative Review Centered on the Etiology, Diagnostics, Prevention, and Therapeutic Options"

_children, 2022, doi:10.3390/children10010083_

Round 1

Reviewer 1 Report

Dear author, 

Your article The entity of cerebellar mutism syndrome: a narrative review centered on the etiology, diagnostics, prevention, and therapeutic options is interesting bur you need some options to improve:

- Design of review (key worlds, year of searching, database for searching PubMed and etc) 

- Should include inclusion and exclusion criteria 

Decision minor revision

Author Response

Dear Reviewer,

Thank you for your valuable comments.

We have performed several revisions to our manuscript, as requested.

You have mentioned that the design of the review should be better delineated. All relevant points (key worlds, year of searching, database for searching PubMed) are addressed in the revised version of our manuscript.

You have suggested that inclusion and exclusion criteria should be included in our manuscript. All relevant supplementary information is added to the revised version of our manuscript.

Reviewer 2 Report

1- The English language usage is unsatisfactory, especially with respect to use of non-technical words like operation for surgery.

2- Certain short forms are not expanded at their first usage e.g., POP-CMS.

3- As it is the authors who are writing this review, they need to take ownership of the statements made, eventhough they may provide references for the same. Writing "as per a recent study..." and then giving the definition followed by 6 references, undermines your contribution to the work.

4- Occupational/speech therapy and behavioral interventions as modalities of management of CMS may be incorporated in the discussion. A table with major findings of previous case series/studies on CMS woukld be a good addition. 

Author Response

Dear Reviewer,

Thank you for your valuable comments regarding our manuscript. We have added several corrections and sections, as requested.

You have mentioned that  English language usage is unsatisfactory. We have performed extensive language and Grammar editting, as requested.

You have stated that certain short forms are not expanded at their first usage e.g., POP-CMS. We have performed all necassary corrections.

You have mentioned that we need to take ownership of the statements made, eventhough we may provide references for the same. We have followed your instructions and performed the neccessary modifications.

You have stated that 'Occupational/speech therapy and behavioral interventions as modalities of management of CMS may be incorporated in the discussion. A table with major findings of previous case series/studies on CMS woukld be a good addition. ' We have included niew sections and a relevant table, following your suggestion.

Round 2

Reviewer 2 Report

None